# Case Series of 11 *CDH1* Families (47 Carriers) Including Incidental Findings, Signet Ring Cell Colon Cancer and Review of the Literature

**DOI:** 10.3390/genes14091677

**Published:** 2023-08-25

**Authors:** Mathis Lepage, Nancy Uhrhammer, Maud Privat, Flora Ponelle-Chachuat, Myriam Kossai, Julien Scanzi, Zangbéwendé Guy Ouedraogo, Mathilde Gay-Bellile, Yannick Bidet, Mathias Cavaillé

**Affiliations:** 1Département d’Oncogénétique, Centre Jean Perrin, 63011 Clermont-Ferrand, France; nancy.uhrhammer@clermont.unicancer.fr (N.U.); maud.privat@clermont.unicancer.fr (M.P.); flora.ponelle-chachuat@clermont.unicancer.fr (F.P.-C.); mathilde.gay-bellile@clermont.unicancer.fr (M.G.-B.); mathias.cavaille@clermont.unicancer.fr (M.C.); 2INSERM, U1240 Imagerie Moléculaire et Stratégies Théranostiques, Université Clermont Auvergne, 63000 Clermont-Ferrand, France; myriam.kossai@clermont.unicancer.fr (M.K.); yannick.bidet@uca.fr (Y.B.); 3Department of Pathology and Molecular Pathology, Centre Jean Perrin, 63011 Clermont-Ferrand, France; 4Centre Hospitalier de Thiers, 63300 Thiers, France; jscanzi@chu-clermontferrand.fr; 5Service de Biochimie et Génétique Moléculaire, CHU Clermont-Ferrand, 63000 Clermont-Ferrand, France; zgouedraogo@chu-clermontferrand.fr; 6CNRS, INSERM, iGReD, Université Clermont Auvergne, 63001 Clermont-Ferrand, France

**Keywords:** HDGC, signet ring cell carcinoma, *CDH1*, E-cadherin, incidental finding, endoscopic finding, prophylactic gastrectomy, HBOC

## Abstract

Germline pathogenic variants in E-cadherin (*CDH1*) confer high risk of developing lobular breast cancer and diffuse gastric cancer (DGC). The cumulative risk of DGC in *CDH1* carriers has been recently reassessed (from 40–83% by age 80 to 25–42%) and varies according to the presence and number of gastric cancers in the family. As there is no accurate estimate of the risk of gastric cancer in families without DGC, the International Gastric Cancer Linkage Consortium recommendation is not straightforward: prophylactic gastrectomy or endoscopic surveillance should be proposed for these families. The inclusion of *CDH1* in constitutional gene panels for hereditary breast and ovarian cancer and for gastrointestinal cancers, recommended by the French Genetic and Cancer Consortium in 2018 and 2020, leads to the identification of families with lobular cancer without DGC but also to incidental findings of pathogenic variants. Management of *CDH1* carriers in case of incidental findings is complex and causes dilemmas for both patients and providers. We report eleven families (47 *CDH1* carriers) from our oncogenetic department specialized in breast and ovarian cancer, including four incidental findings. We confirmed that six families did not have diffuse gastric cancer in their medical records. We discuss the management of the risk of diffuse gastric cancer in Hereditary Lobular Breast Cancer (HLBC) through a family of 11 *CDH1* carriers where foci were identified in endoscopic surveillance. We also report a new colon signet ring cancer case in a *CDH1* carrier, a rare aggressive cancer included in *CDH1*-related malignancies.

## 1. Introduction

Hereditary diffuse gastric cancer syndrome (HDGC) is characterized by a high prevalence of diffuse gastric cancer (DGC) and lobular breast cancer (LBC). It is associated with the *CDH1* gene that encodes for the E-cadherin protein, a trans-membrane glycoprotein, which plays a major role in cell–cell adhesion and tissue integrity, performing critical mechanical and signalling functions in epithelial cells [1]. As a tumor suppressor, it is downregulated among the initiating steps of the epithelial-mesenchymal transition, resulting in cellular plasticity and a migratory phenotype required for metastatic disease [2]. In embryogenesis, E-cadherin is the first adhesion molecule expressed at the eight-cell stage and it is highly expressed during critical lip and palate development stages. *CDH1* germline mutations have been also detected in patients with syndromic and non-syndromic cleft lip/palate and more recently in blepharocheilodontic syndrome [3,4,5].

*CDH1*-driven gastric cancers present as signet ring cell carcinoma (SRCC) and readily metastasize before forming large primary lesions, typically presenting at late stages upon detection (43% of SRCC are detected at a distant or metastatic stage vs. 37% for other gastric adenocarcinomas) [6]. Overall, the cumulative incidence of gastric cancer by age 80 years for pathogenic *CDH1* variant carriers is now estimated at 37 to 42% for men and 25 to 33% for women, about half of the initial estimation described before the advent of high-throughput sequencing. Pathogenic variants in *CDH1* also predispose women to develop LBC, with a lifetime risk of about 42 to 55% [7,8].

The distinction between hereditary diffuse gastric cancer (HDGC) and hereditary lobular breast cancer (HLBC) made by the International Gastric Cancer Linkage Consortium (IGCLC) in 2020 acknowledges the likelihood that not all families with pathogenic *CDH1* variants are equally at risk of DGC. While prophylactic gastrectomy is recommended for pathogenic *CDH1* variant carriers from families with gastric cancer between 20 and 30 years of age, the recommendation for families without DGC is not straightforward. When there is no familial history of DCG, yearly endoscopic surveillance including random biopsy should be offered to these patients, but prophylactic total gastrectomy should also be considered, giving careful attention to the uncertain gastric cancer risk [9]. The limited sensitivity of endoscopic surveillance must be clearly explained to the patient.

The inclusion of *CDH1* in HBOC and gastrointestinal gene panels, recommended by the French Genetic and Cancer Consortium, has led to the identification of increasing numbers of families with lobular cancer without DGC but also in incidental findings [10,11]. Thus, while multigene panel sequencing allows for rapid analysis of multiple cancer susceptibility genes, is cost-effective, and leads to incremental genetic findings, it also causes dilemmas for both patients and providers in case of incidental finding of *CDH1* mutation [12,13].

In this report, through the presentation of eleven families, we report our experience in the management of *CDH1* carriers, especially in families without DCG (HLBC and incidental findings). We also report a new signet ring cell colorectal cancer case in a *CDH1* carrier and discuss *CDH1*-related cancer.

## 2. Materials and Methods

### 2.1. Population Included

All families who had a pathogenic or likely pathogenic (P/LP) variant in *CDH1* (NM_004360.5) were recruited from our oncogenetic department from 2014 to 2022, either through *CDH1*-oriented analysis or multigene panel testing performed in patients with suspected hereditary predisposition to cancer (2753 analyses). Each patient signed informed consent for genetic diagnosis of hereditary disease and for research. Patient follow-up was only available for patients who agreed to be included in our oncogenetic clinical follow-up program.

*CDH1*-oriented analysis was performed if the IGCLC criteria were met (Table 1). These patients received specific pre-test counselling on *CDH1*. Multigene panel testing included *CDH1* as recommended by the French Genetic and Cancer Consortium (GGC) in two indications, and without specific pre-test counselling on *CDH1:*In patients with suspected HBOC syndrome from 2018 (2453 analyses), defined as patients with Grade A or B according to French Society of Predictive Medicine criteria [14]. The HBOC panel included *BRCA1, BRCA2, PALB2, TP53, CDH1, PTEN, RAD51C, RAD51D, MLH1, MSH2, MSH6, PMS2* and *EPCAM.*In patients with suspected hereditary predisposition to digestive cancer from 2019 (300 analyses) [15]. Gastrointestinal panel included *APC, BMPR1A, CDH1, EPCAM, MLH1, MSH2, MSH6, MUTYH, PMS2, POLD1, POLE, PTEN, SMAD4* and *STK11*.

### 2.2. Molecular Analysis

Index cases were explored using next-generation sequencing, from EDTA peripheral blood. DNA was extracted from blood using manual extraction (QIAamp DNA Blood maxikit) or automated QIAsymphony extraction (Qiagene, Hilden, Germany). DNA was fragmented either by sanitation using a Bioruptor Pico Instrument (Diagenode, Liège, Belgium) or enzymatically as part of the Kapa library kit. Kapa HyperPrep(+) library preparation and SeqCap EZ Choice probes and reagents (Roche, Bâle, Switzerland) were used for library preparation and capture. Quality of fragmentation, library and capture were controlled using a Bioanalyzer 2100 or a Tapestation 4150 instrument (Agilent, Santa Clara, CA, USA). Sequencing was performed using 300-cycle Illumina kits on Miseq or NextSeq 550 Instruments (Illumina, San Diego, CA, USA). All steps were performed following providers’ guidelines, aiming for a minimal depth of 30X. Variants were classified according to the American College of Medical Genetics (ACMG) recommendations and the most recent guidelines on *CDH1* variant classification, aided by the French National Database of variants [16,17]. Confirmation of P/LP variants in index cases and targeted analysis of family relatives were performed using BigDye terminator kit 3.1 (Applied Biosystems, Waltham, MA, USA) reagents for Sanger sequencing. CNV were confirmed by quantitative multiplex polymerase chain reaction of short fluorescent fragments (QMPSF) analysis. Both techniques ran on a 3500xl instrument (Applied Biosystems, Waltham, MA, USA).

## 3. Results

A total of 47 patients (23 men and 24 women) with germline *CDH1* P/LP variants in 11 families were recruited. They had no other known P/LP variant in genes explored by our panel. Six *CDH1* carriers had diffuse gastric cancer (five families) and 11 women had lobular breast cancer. The mean age at diagnosis was 42 (+/−20) years for diffuse gastric cancer, and 54 (+/−19) years for lobular breast cancer. Both ages are younger than reported in the general population, respectively, 63 and 59 years [18,19]. Seven *CDH1* carriers had a prophylactic gastrectomy and foci were observed in 57% of cases. A colorectal SRCC was observed at age 30 in an individual with a pathogenic variant.

Demographic and clinical features for each family are summarised in Table 2. P/LP variants included six premature termination variants, three large deletions and two splice variants. All families’ pedigrees are available in the additional data (Appendix A).

Six families out of eleven met the IGCLC criteria (five HDGC and one HLBC). In one family, all that was missing was a pathological report of breast cancer. The last four families were incidental findings that did not meet IGCLC criteria, three from HBOC panels analyzed for nonspecific breast cancer and one from a gastrointestinal panel in the context of polyposis. Lerner et al. proposed in 2022 less restrictive criteria to test *CDH1* [20]. The Yale criteria include all cases with National Comprehensive Cancer Network (NCCN) HBOC criteria or patients with DGC at any age and patients with family history of two or more cases of gastric cancer in first-degree or second-degree relatives when at least one is confirmed to be DGC or diagnosed at age 50 or younger. All our families except one fulfilled the Yale criteria which seem to be more sensitive.

(A)Gastric and/or lobular breast cancer families.

Family 1:

Three sisters developed lobular breast cancer at 35, 45 and 54. One of their daughters had breast cancer at age 40, a son was diagnosed with DCG at 40, and a nephew had colon signet ring cell carcinoma at age 30. The pathogenic variant c.2386delC p.(Arg796Glyfs*20) of *CDH1* was identified in five of these six patients (one of the sisters who did not undergo genetic testing). Two sons of the sisters underwent prophylactic gastrectomy where multiple foci were found, some of which invaded the submucosa (pT1a). In one case, prophylactic gastrectomy was complicated by hemorrhagic shock with rupture of the splenic artery, the common hepatic artery and the duodenal artery, pancreatic necrosis and a duodenal fistula. The familial investigation is ongoing, with the identification of two new carriers in grandchildren.

Family 2:

A female patient was referred to our cancer genetics clinic for lobular breast cancer at 54 years of age. A family history of breast cancer and DGC was noted, affecting her two sisters at 49 and 40 years. The genetic analysis of *CDH1* revealed the germline pathogenic variant c.469del; p.(Val157Leufs*58). The proband’s father did not carry the *CDH1* variant, and her mother (who refused genetic testing) was free from cancer at 80. Her daughter, and her sister affected by DGC, were also identified as *CDH1* carriers. The proband and her daughter initially refused prophylactic gastrectomy and they were followed by annual endoscopy. The proband’s daughter had a positive sample biopsy for signet ring gastric cancer identified 3 years later (33 years old). She underwent total gastrectomy with a subsequent diagnosis of SRGC (pTis pN0).

Family 3:

Two brothers visited our hospital for DGC at 39 and 43. Their paternal uncle was also diagnosed with DGC at 39 years of age and their paternal grandmother had bilateral lobular breast cancer at 70. Genetic testing revealed *CDH1* c.1795delA p.(Thr599Leufs*13) germline pathogenic variant for the two patients. Both patients died rapidly after the diagnosis of DGC. Two nephews decided to undergo *CDH1* genetic testing for cancer prevention; both were negative.

Family 4:

A woman was referred to our hospital for lobular breast cancer at 45. Her father died from gastric cancer (no pathology report available). An aunt of her father and the aunt’s two daughters reported lobular breast cancer, two of which were metachronous bilateral lobular breast cancer. Breast cancer occurred between the ages of 42 and 79. The germline pathogenic *CDH1* variant c.2398del; p.(Arg800Alafs*16) was detected in these patients. The proband had a prophylactic gastrectomy where no foci were identified.

Family 5:

A woman was referred to a hospital for a DGC at 40. There was no family history of cancer among first-degree relatives. Her grandmother had breast and colorectal cancer. A germline pathogenic variant was identified c.1488_1494del; p.(Glu497Leufs*23). One of her sisters was also a *CDH1* carrier. She refused prophylactic gastrectomy and after a first endoscopy, she dropped follow-up.

Family 6:

A woman with bilateral synchronous lobular breast cancer at 42 was addressed to our cancer genetics clinic. Her mother, her maternal aunt and two maternal female cousins had lobular breast cancer at 50, 53, 60 and 63, one of which was bilateral. The germline pathogenic *CDH1* variant c.283C>T; p.(Gln95*) was detected. Overall, 10 *CDH1* carriers were identified in this family and there was no history of invasive DGC cancer. *CDH1* carriers had annual endoscopic surveillance based on the Cambridge protocol: a positive biopsy with SRCC invading gastric mucosa was observed in a nephew of the proband. The nephew had a final diagnosis of PT1a gastric cancer.

Family 7:

The proband had lobular breast cancer at 46 years of age, and her sister a bilateral breast cancer at 40 (no pathology report available). Her mother also had lobular breast cancer at 60 and recurrent breast cancer at 70. A germline deletion of the exon 16 was identified c.(2439+1_2440-1)_(*1_?)del; p.? in the proband. Because of the loss of the last 69 amino acids of the cytoplasmic domain, the variant classification was confirmed by the French laboratory expert group as probably pathogenic. This variant was not detected in her father. To note, her sister had a prior genetic test, negative for BRCA genes in 2015. The proband chose to undergo annual endoscopic surveillance.
(B)Incidental findings
Family 8:

A woman was referred to our consultation for nonspecific breast cancer at 39 years of age. She had no family history of cancer, except for her grandmother who had lung cancer. A panel for hereditary breast and ovarian cancer including *CDH1* was performed: a large deletion leaving only the first two exons of *CDH1* was detected (c.(163+1_164-1)_(*1_?)del; p.?). The pathogenic deletion was also identified in one of her female cousins. Both chose to have prophylactic gastrectomy, but no foci were identified. One patient had severe complications which required additional intervention. Her father and aunt, who were obligate carriers of *CDH1*, had not developed any cancer at ages 61 and 63.

Family 9:

A woman had a diagnosis of bilateral nonspecific breast cancer at 43 years old. In her family, one of her paternal female cousins had lobular breast cancer, and her paternal aunt a breast cancer at age 45 (no pathology report available). Her mother was adopted but was free from cancer at 70 years old. A panel for hereditary breast and ovarian cancer including *CDH1* was performed: a germline pathogenic splice variant of *CDH1* c.1901C>T; r.1900_1936del; p.(Ala634Profs*7) was detected. A targeted *CDH1* test and a panel for hereditary breast and ovarian cancer did not find any pathogenic variant in her female cousin with lobular breast cancer. The proband had a first endoscopy without foci and she will have a prophylactic gastrectomy soon.

Family 10:

The proband had nonspecific breast cancer at 58. She has four brothers and four sisters. Two sisters had breast cancer at ages 54 and 60 (no pathology report available). There was no other familial history of breast or gastric cancer in her first- and second-degree relatives. Of note, her maternal grandmother died at 33 from uterine cancer in 1931 (no pathology report available). An out-of-frame pathogenic deletion of exons 11 and 12 in CDH1, c.(1320+1_1321-1)_(1711+1_1712-1)del), has been identified. The familial investigation is ongoing, and the proband will have a primary endoscopic work-up soon.

Family 11:

A man presented clinical polyposis at 75 (>25 adenomatous polyps). His son had also a polyposis and developed colorectal cancer at age 30. His paternal grandmother had breast cancer (no pathology report available). The proband had a gastrointestinal panel recommended. A germline pathogenic splice variant of *CDH1* c.2195G>A was identified. Previous RT-PCR analysis and minigene assay showed that this alteration activates a cryptic acceptor site and leads to alternate splicing and deletion of 32 base pairs at the start of exon 14 [21]. There was no other familial history of breast or gastric cancer. Familial investigation and initial endoscopic for the proband are in progress.

## 4. Discussion

We report 11 families with diverse presentations, including four incidental findings, two families with only LBC, and five families with at least one DGC. Similar proportions were observed in a larger cohort (113 probands): 36% of families had mixed gastric/breast cancer, 36% had breast cancer in the absence of gastric cancer, 16% had gastric cancer in the absence of breast cancer and the remaining 12% of families had no gastric or breast cancer (in total, 48% of the families had no gastric cancer) [7].

Until recently, prophylactic gastrectomy was recommended for all *CDH1* carriers, as the risk of gastric cancer was estimated to be over 70% for all families. In 2020, the IGCLC recognized the variability in gastric cancer risk between families. Risk of gastric cancer was estimated at 64% for men and 47% for women when three or more DGC are present in the family and 27% for men and 24% for women when two or fewer DGC are observed [8]. There is no estimation of DGC in a large cohort of HLBC. The risk should be probably lower, although advanced SRCC have been found in several patients who had a prophylactic gastroscopy despite no family history [22]. Furthermore, the prevalence of foci in the HLBC and HDGC groups is similar [23]. Given the uncertain risk of DGC in these families, they should be offered a choice between prophylactic gastrectomy and endoscopic surveillance after explaining the risks and benefits of each procedure.

The sensitivity of endoscopic surveillance is limited. Using the Cambridge endoscopy protocol, two prospective studies observed foci in 61.1% and 67.3% of cases [24,25]. However, precursor lesions (pT1a) and/or invasive carcinoma foci are identified in more than 95% of prophylactic gastrectomy in *CDH1* carriers [26]. The improvement of the foci detection rate with the Cambridge protocol is also controversial in the literature: Benesh et al. reported sensitivity of 20 to 28% for endoscopic surveillance, with no statistical improvement via the Cambridge Protocol [27]. Furthermore, the clinical relevance of superficial SRCC in endoscopic biopsies is questionable, since these superficial lesions can display very indolent behaviour [28,29]. If 95% of *CDH1* carriers have at least one signet ring cell carcinoma lesion in the total gastrectomy, and the cumulative risk of gastric cancer at 80 years old is 20 to 42%, more than half of CDH1 carriers with foci will not develop DGC.

For these reasons, a shift in the paradigm of endoscopic surveillance is discussed: the aim is not to detect all precursor lesions but to detect abnormal lesions that tend to infiltrate deeper toward the submucosa. Recognising the lesion limited to the mucosa which tends to progress is probably the biggest challenge of endoscopic surveillance for these patients. A key feature proposed by Asif et al. is the presence of ulceration, whereas Van der Post et al. proposed a new categorisation for endoscopic findings, with pT1 with atypia defined by the presence of deeper infiltration towards or in the submucosa; mixture with smaller or pleomorphic and eosinophilic cells; atypical cells not restricted to the base of the lesion; some inflammatory or stromal reaction; or increased proliferation [30,31]. A prospective study found two cancers at an advanced stage (pT3) in 120 patients in the endoscopic surveillance arm but did not observe any invasive gastric cancer in prophylactic gastrectomy specimens from patients (98) who did not have any concerning findings in their endoscopy [29]. From the review of three other published studies, Dardene et al. find that, of 147 prophylactic gastrectomies in asymptomatic carriers, seven had advanced gastric cancer, only one of which was not seen by the endoscopy [32]. In one of our HBLC families, all 12 carriers were followed by endoscopic surveillance. A precursor lesion without atypical finding was identified in an asymptomatic carrier at 26 years old. However, as there was no guarantee that this lesion would not progress and the prognosis of advanced DGC is poor (5-year overall survival rate of 10–20%), the patient chose to undergo gastrectomy. We still consider this family as an HLBC and did not recommend prophylactic gastrectomy to relatives. Although progress has been observed in endoscopic surveillance, it is still difficult to postpone gastrectomy when superficial lesions are identified, even though these lesions have an indolent profile. Endoscopic surveillance must also be performed in an expert center.

In the six families without DGC, four carriers chose to undergo prophylactic gastrectomy (three have been done and one is pending). Foci were detected only in the patient mentioned above where SRCC was detected by endoscopic surveillance. Overall, seven prophylactic gastrectomies were performed in the 11 families, and foci were identified in four cases (57%). Our low rate of foci discovery on prophylactic gastrectomy could be explained by the absence of exploration of the entire mucosa. Without a specific protocol, foci are detected only in 62.5% of prophylactic gastrectomies [25].

In contrast to endoscopic surveillance, prophylactic gastrectomy reduces the risk of developing HDGC to virtually 0%, but it has a 100% risk of side effects. The most frequent include early and late dumping syndrome, malabsorption, weight loss and postprandial fullness. In a prospective study of gastrectomy in *CDH1* carriers, the rate of complications after surgery varied from 0% to 27% requiring additional intervention, and 2.5% mortality [33,34,35]. In our cohort, two patients had severe complications which required additional surgery (haemorrhagic shock, pancreatic necrosis, duodenal fistula). Long-term outcomes specific to patients with prophylactic gastrectomy at a young age are not well established. A multidisciplinary team approach is needed for long-term monitoring, which should include supplementation of multivitamins and minerals, management of micronutrient supplementation and iron deficiency, annual blood testing and prevention of reduced bone mineral density [36].

In contrast to the risk of gastric cancer, the risk of lobular breast cancer has not changed significantly since the first description (cumulative risk at 80 years estimated at 40–55%). The diagnosis of lobular breast cancer is challenging, as these tumors are associated with the loss of cell adhesion molecule E-cadherin, leading to cells with a discohesive morphology. The French guidelines for female *CDH1* carriers recommend annual breast MRI between 30 and 65 years of age, while the IGCLC guidelines recommend breast MRI between 30 and 50 years of age (potentially longer). In our eleven families, half of the lobular breast cancers (8/15) in *CDH1* carriers occurred after 50 years (mean age 54 years old (SD 19). Roberts et al. estimated the mean (SD) age at diagnosis of 48.2 (10.9) years [8]. The sensitivity of mammography, particularly related to breast density, is lower for lobular breast cancer (57% to 81%) than for nonspecific breast cancer (63% to 98%) [37]. MRI should be thus performed even after 50 years of age, as more than half of the women have dense breasts in their 50s and the risk of breast cancer is still high [38]. Bilateral risk-reducing mastectomy can be considered for *CDH1* women carriers.

The inclusion of *CDH1* in multigene panel testing (in France, HBOC and GI panels) has redefined the cancer risk but it has led to the identification of incidental mutations. In our study, incidental findings in three families were secondary to HBOC (0.1% of our HBOC analyses) and a gastrointestinal panel in one family (0.3% of our GI panels). These patients did not receive specific pre-test counselling concerning *CDH1* as there was no significant familial history. The lack of such a counselling left these families in an unexpected situation, fraught with emotional and physical ramifications for all involved family members, as well as difficult decisions regarding whether to pursue risk-reducing total gastrectomy. In two families, the probands chose to undergo prophylactic gastrectomy. One of the probands was overweight (IMC 42.45), which influenced her decision although there were psychological repercussions for her and her family. In the other two families, the first endoscopic workups are ongoing. Germline pathogenic splice variant c.1901C>T (r.1900_1936del; p. Ala634Profs*7) was identified in one of these families (the proband had a nonspecific breast cancer at 39 years old and there was no other familial history of cancer, but her mother was adopted). This splice variant was recently reported as a founder variant in the Portuguese population with low penetrance (frequency of DGC and LBC was 18.9% and 19.4% in 58 carriers of *CDH1* c.1901C>T) [39]. This reduced penetrance may explain the lack of familial cancer related to *CDH1*, although to date there is no validated genotype–phenotype correlation to adapt follow-up recommendations [40].

For the three incidental findings after HBOC panel analysis, all probands had nonspecific breast cancer. Nonspecific breast cancer is not included in the *CDH1* spectrum by the IGCLCC, although in a cohort of 25 female *CDH1* carriers who reported breast cancer, 32% had invasive ductal carcinoma [8]. Lowstuter et al. showed that when E-cadherin immunohistochemical staining was retrospectively completed on three invasive ductal tumors from women with germline *CDH1* pathogenic variants, two-thirds were confirmed as ductal, whereas one-third were reclassified as lobular, based on the lack of E-cadherin expression [41]. In our three incidental findings from the HBOC panel (families 8, 9 and 10), an expert pathological review of tumor tissues confirmed the initial classification of nonspecific breast cancer. Immunohistochemical staining of E-cadherin was normal in two cases (families 8 and 9) and was reduced in one nonspecific breast cancer (family 10). Abnormal staining of E-cadherin can occur in nonspecific breast cancer, more frequently in the advanced or metastatic stages (our patient had a pT4 breast cancer). Although we report nonspecific breast cancer in *CDH1* carriers, there is no evidence of a casual effect.

We also observed a colorectal signet ring cell carcinoma in a *CDH1* carrier at 30 years old. Colorectal SRCC is a rare adenocarcinoma subtype (1–2.4% of all colorectal cancer) with a poor prognosis overall, associated with somatic loss of E-cadherin. Prior case reports suggested that an association between SRCC and *CDH1* germline carriers exists but there was no evidence of increased risk in a larger cohort [42,43,44]. Recently, histology-specific enrichment analysis identified an association between DGC and SRCC [27]. Adib et al. found significant enrichment of P/LP variants of *CDH1* in patients with SRCC [45]. SRCCs are probably included in *CDH1*-related cancer although the risk seems low. Colonoscopic surveillance should be considered in *CDH1* carriers with familial history of SRCC.

## 5. Conclusions

In summary, risk stratification and management of *CDH1* carriers are difficult, as not all families are equally at risk of DGC. In this context, incidental findings of *CDH1*, which are more frequent since the inclusion of *CDH1* in multigene panel testing, cause dilemmas for both patients and providers. We believe that when there is no DCG in the family, after a clear explanation of the benefits and risks of each procedure, the patient’s will is paramount in the choice of prophylactic gastrectomy or endoscopic surveillance. Breast MRI should be performed even after the age of 50, as the risk of breast cancer is still high and the sensitivity of mammography not optimal. Lastly, the spectrum of *CDH1*-related cancers probably not only includes DGC and lobular breast cancer but also signet ring cell colorectal cancer.

## Figures and Tables

**Table 1 genes-14-01677-t001:** Hereditary diffuse gastric cancer (HDGC) genetic testing criteria from IGCLC 2020.

Family * Criteria	Individual Criteria
(1) ≥ 2 cases of gastric cancer in family regardless of age, with at least one DGC (2) ≥ 1 case of DGC at any age, and ≥ 1 case of lobular breast cancer at age < 70 years, in different family members (3) ≥ 2 cases of lobular breast cancer in family members < 50 years of age	(1) DGC at age < 50 years (2) DGC at any age in individuals of Māori ethnicity (3) DGC at any age in individuals with a personal or family history (first-degree relative) of cleft lip or cleft palate (4) History of DGC and lobular breast cancer, both diagnosed at age < 70 years (5) Bilateral lobular breast cancer, diagnosed at age < 70 years (6) Gastric in situ signet ring cells or pagetoid spread of signet ring cells in individuals < 50 years of age

* Family members must be first-degree or second-degree blood relatives.

**Table 2 genes-14-01677-t002:** Clinical characteristics of the 11 families.

Families	Variant	Number of *CDH1* Carriers	DGC from *CDH1* Carriers	Breast Cancer from *CDH1* Carriers	DGC and Breast Cancer from *CDH1* Untested Relatives	Prophylactic Gastrectomy in *CDH1* Carriers	*CDH1* Carriers without DGC or Breast Cancer	IGCLC Criteria	Yale Criteria	Others Features
1	c.2386delC; p.(Arg796Glyfs*20)	n = 12 (8M/3W)	n = 1 (40)	n = 3 LBC (35, 40, 54)	n = 1 NSP (45)	2 PG = 2 foci	6 (30, 36, 77)	present	present	Signet ring cell colorectal cancer in *CDH1* carrier
2	c.469delG; p.(Val157Leufs*58)	n= 5 (2M/3W)	n = 1	n = 1 LBC (54)	n = 1 BC (49) (no pathology report)	1 PG = PTis	1 (>70)	present	present	
3	c.1795delA; p.(Thr599Leufs*13)	n = 2 (M)	n = 2 (39, 43)	n = 0	n = 1 DCG n= 1 bilateral LBC 70	No PG	1 (60)	present	present	
4	c.2398delC; p.(Arg800Alafs*16)	n = 6 (1M/6W)	n = 1 (63) (no pathology report)	n = 4 2 LBC (45, 60) 1 bilateral LBC (41, 60) 1 bilateral breast cancer LBC and NSP (62, 79)	0	1 PG = no foci	1 (56)	present	present	
5	c.1488_1494del; p.(Glu497Leufs*23)	n = 3 (1M/2W)	n = 1 (42)	n = 0	n = 1 BC (no pathology report)	n = 0	1 (65)	present	present	
6	c.(2439+1_2440-1)_(*1_?)del	n = 1 (W)	n = 0	n = 1 LBC (46)	n = 1 LBC (60) n= 1 bilateral BC (no pathological report) 40	No foci after first endoscopy	0	absent missing one pathological report	present	
7	c.283C>T; p.(Gln95*)	n = 12 (5M/7W)	n = 0	n = 3 1 LBC (60) 2 bilateral LBC (62, 63) (42, 42)	n = 2 LBC (53, 50)	1 PG = pT1a	7 (63, 75, 31, >70)	present	present	
8	c.(163+1_164-1)_(*1_?)del	n = 4 (3M/1W)	n = 0	n = 1 NSP cancer (39)	n = 0	2 PG = 0 foci	3 (32, 61, 63)	absent	present	
9	c.1901C>T; (r.1900_1936del; p. Ala634Profs*7)	n = 1 (W)	n = 0	n = 1 Bilateral NSP (43)	n = 1 (45) BC (no pathology report)	PG intented	0	absent	present	Adopted mother
10	c.(1320+1_1321-1)_(1711+1_1712-1)del	n = 1 (W)	n = 0	n = 1 NSP	n = 3 BC (54, 60) (no pathology report)	Initial endoscopy in progress	0	absent	present	
11	c.2195G>A; p.(Arg732Gln)	n = 1 (M)	n = 0	n = 0	n = 0	Initial endoscopy in progress	0	absent	absent	
Total		n = 47	n =6	n = 15	n = 10 BC/1 DCG	4 foci (7 PG)	18			

LBC: lobular breast cancer; NSP: nonspecific breast cancer; PG: prophylactic gastrectomy; DCG: Diffuse gastric cancer; M: men; W: women; n: number. Five families had DGC and breast cancer (blue). Two families presented only breast cancer (orange). Four families did not present lobular breast cancer or DGC (grey).

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
