# Peer review of "Case Series of 11 CDH1 Families (47 Carriers) Including Incidental Findings, Signet Ring Cell Colon Cancer and Review of the Literature"

_genes, 2023, doi:10.3390/genes14091677_

Round 1
Reviewer 1 Report
In this paper by Lepage et al, the authors report on 11 families of CDH1 pathogenic mutation carriers from a single site high-risk cancer clinic. The families were identified for testing either through the very strict International Gastric Cancer Linkage Consortium criteria or through testing of individuals who qualify for hereditary breast and ovarian cancer (HBOC) and hereditary gastrointestinal cancer criteria according to the current French Genetic and Cancer Consortium guidelines. As expected, this resulted in a mixed pattern of families with different phenotypes that include gastric and lobular breast cancer, non-specific breast cancer only, and even an early-onset colorectal cancer.
The paper reinforces the recent findings that have challenged knowledge on CDH1 clinical phenotypes, and the reported families illustrate these differences. Data seems accurately reported. Families are mostly well described though some important pathology data is missing in families 8-10. Background and discussion are very well put together and overall contribute to bring perspective into the challenges of care management of CDH1 mutation carriers.
Specific comments:
1. I would like for the authors to explain what the HBOC and hereditary gastrointestinal cancer criteria are according to the French guidelines. Are they any different from the ones used in the US, particularly the NCCN guidelines?
2. Could the authors obtain slides from breast cancer cases in families 8, 9 and 10 and have an expert Pathologist review them?
3. In family 11, did the individual who developed colorectal cancer at 30 have polyposis like his father?
Reviewer 2 Report
This is a description of a limited series of CDH1 carriers, but it brings value to the field of CDH1.
The reviewer has some questions that need to be addressed and a suggestion that would improve the manuscript.
1- "Of the 11 families, six families respected the IGCLC criteria, and in one family was 1 only missing a breast cancer pathology report (5 HDGC and 2 HLBC)."
Regarding the last family. Can the authors clarify whether that family fulfilled the criteria?
2-Did the families diagnosed as incidental findings fulfill the criteria?
3- Add one column in the table describing what families fulfill the criteria.
4-Lerner BA, et al. (PMID: 35078942) proposed the Yale criteria which identifies cases using a less cumbersome set of criteria than the IGCLC 2020. It would be interesting to include the use of this criteria in this series of cases to evaluate its efficiency and compare with the IGCLC 2020.
Page 4, line 1, this sentence needs revision. respected is not the best word, fulfilled would work better and the second sentence is not clear enough.
"A female patient was referred to our oncogenetic consultation for lobular breast cancer at 54 years of age."
"A woman with bilateral synchronous lobular breast cancer at 42 was addressed to our oncogenetic consultation."
These sentences would benefit from revision. Maybe use the term, referred to our cancer genetics clinic
References are all messed up, needs formatting
Round 2
Reviewer 2 Report
The authors have addressed the reviewer's previous suggestions.
- All our families except one respected the Yales criteria which seem more sensitive.
This sentence should be: “All our families fulfilled the Yale criteria which seem to be more sensitive”
- Family 11:
A man presented clinical polyposis at 75 (> 25 adenomatous polyps). His son had also a polyposis and developed colorectal cancer at age 30.
This sentence should be: His son also had polyposis and developed colorectal cancer at age 30.
- The Reference section looks better, but it should be all with the same font.
Author Response
Dear reviewer ,
The two sentences have been corrected :
- All our families except one fulfilled the Yale criteria which seem to be more sensitive Page 4 L138 (Table 2 : family 11)
- His son had also a polyposis and developed colorectal cancer at age 30 Page 7 L 90
The font reference has been changed.
Thank you
Lepage Mathis